# Differences in prescribed medicinal cannabis use by cannabinoid product composition: Findings from the cannabis as medicine survey 2020 (CAMS-20) Australia-wide study

Benjamin T. Trevitt[1,2]*, Sasha Bailey[3], Llewellyn Mills[1,4,5], Thomas R. Arkell[6], Anastasia Suraev[7,8,9], Iain S. McGregor[7,8,9ʘ], Nicholas Lintzeris[1,4,5ʘ]

**1** Drug and Alcohol Services, South Eastern Sydney Local Health District, Sydney, NSW, Australia, **2** School of Public Health and Community Medicine, UNSW Sydney, Sydney, Australia, **3** Faculty of Medicine and Health, The Matilda Centre for Research in Mental Health and Substance Use, The University of Sydney, Camperdown, Australia, **4** Speciality Addiction Medicine, Faculty Medicine and Health, University of Sydney, Camperdown, NSW, Australia, **5** Drug and Alcohol Clinical Research and Improvement Network, St Leonards, NSW, Australia, **6** Centre for Human Psychopharmacology, Swinburne University of Technology, Hawthorn, VIC, Australia, **7** The University of Sydney, Lambert Initiative for Cannabinoid Therapeutics, Sydney, NSW, Australia, **8** The University of Sydney, Faculty of Science, School of Psychology, Sydney, NSW, Australia, **9** The University of Sydney, Brain and Mind Centre, Sydney, NSW, Australia

ʘ These authors contributed equally to this work.
* benjamin.trevitt@health.nsw.gov.au

## Abstract

## Introduction

Prescribed medicinal cannabis (MC) is an increasingly common prescription in Australia for treating pain, anxiety, and sleep disorders. Prescribed MC products generally contain tetra-hydrocannabinol (THC) and/or cannabidiol (CBD) in a variety of dose levels and forms. It is unclear whether THC and CBD products are used by patients with different characteristics and for different conditions.

## Objectives

To examine consumer experiences of using THC- and CBD-containing prescribed MC products to better understand how they are being used within the Australian context.

## Methods

We utilised data collected from an online anonymous cross-sectional survey of individuals (CAMS-20 survey), consisting of Australian residents using cannabis for therapeutic reasons. We focused on a subgroup of participants (N = 546) receiving prescribed MC products. We utilised linear, logistic, and multinomial regression modelling to analyse responses to survey questions based on the cannabinoid profile of the prescribed product.

**Funding:** The authors received no specific funding for this work.

**Competing interests:** Professor Nicholas Lintzeris reports receiving grants from the Australian National Health and Medical Research Council (NHMRC) during the conduct of the study and research grants from Camurus and Indivior – all for unrelated work. Professor Ian McGregor reports receiving grants from Lambert Initiative for Cannabinoid Therapeutics and from NHMRC during the study, but for unrelated projects. Professor Ian McGregor also has patents to WO2018107216A1, WO2017004674A1 and WO2011038451A1 issued and licensed, as well as patents to AU2020050941, AU2019903299, AU2017904438, AU2017904072 and AU2018901971 pending, and a patent WO2019227167 and WO2019071302 issued. Dr Thomas Arkell reports receiving grants from Swinburne University of Technology and the Victorian Department of Health for unrelated projects. No other authors report conflicts of interest. This does not alter our adherence to PLOS ONE policies on sharing data and materials.

## Results

Participants prescribed THC-dominant MC products were statistically more likely to be younger, male, and to prefer inhaled routes of administration than participants using CBD-dominant products who were older, female, and preferred oral routes of administration. Pain and mental health were the most common reasons for all types of prescribed MC, but were more likely to be treated with THC than CBD despite the significantly higher risk of mild to severe drowsiness, dry mouth and eye irritation. Consumer reported effectiveness of prescribed MC was very positive, particularly for THC-containing products. Consumers on opioids and antipsychotics were statistically more likely to be prescribed THC-containing products than products containing CBD only, despite the greater risk of impairment.

## Conclusions

**This Australia-wide study found** clear differences in consumer-reported experiences of prescribed THC- and CBD-containing products. Current prescriptions of these products do not always align with relevant clinical guidance. Educating prescribers around cannabinoid products is essential to ensure optimal prescribing practices and to prevent avoidable drug side effects and interactions.

## Background

There is a global trend toward the legal prescription of cannabinoid-based products for medicinal purposes [1], i.e., prescribed medical cannabis (MC). This is epitomised by the World Health Organisation's 2019 recommendation that cannabis and cannabis resin be deleted from Schedule 4 of the Single Convention on Narcotic Drugs, recognising the emerging therapeutic role of MC [2]. It is also illustrated by the rapid growth of MC prescribing in Australia where, since November 2016, medical practitioners have been able to legally prescribe cannabinoids as unregistered medicines using the compassionate access regulatory pathways (Special Access (SAS) and Authorised Prescriber Schemes) [3]. Clinicians have access to guidelines and educational materials to assist them in making evidence-based decisions in terms of prescribing MC (e.g. prescribing MC for chronic pain or palliative care), however in practice are permitted to prescribe MC for any clinical reason provided they can provide justification to the regulatory body based on available evidence [3]. Using cannabis in Australia that has not been prescribed remains illegal. Data provided by the Therapeutic Goods Administration (TGA), the Australian federal regulator responsible for accessing, monitoring and evaluating products defined as therapeutic goods [4], show that there had been only around 30,000 Special Access Scheme-B (SAS-B) approvals to prescribe MC by the end of 2019 [5]. At the time of writing (March 2023) however, almost 350,000 SAS-B approvals had been issued by the TGA involving around 5,000 medical professionals [6]. In an Australia-wide online survey of medical cannabis consumers, the Cannabis as Medicine Survey 2020 (CAMS-20), 601 (37.6%) of 1600 participants reported using prescribed MC in the past year [3], with the remaining majority using illicitly sourced cannabis products only for medical reasons. The proportion of prescribed MC (as opposed to illicit MC) consumers is rapidly increasing [7] as jurisdictions streamline regulatory processes and clinicians' safety concerns are addressed and stigma surrounding use is reduced [8]. However, current MC prescribing practices often fail to follow guidance provided by the TGA or involve conditions for which no guidance is available [9].

Although the TGA regulates access to prescribed MC, most products have not been formally approved or assessed by the TGA for efficacy, safety, or quality [10].

Two major cannabinoid components are found in MC products: tetrahydrocannabinol (THC) and/or cannabidiol (CBD) [1]. THC and CBD have strikingly different pharmacological properties and, accordingly, products enriched in one or the other have distinct, although somewhat overlapping, applications [11]. CBD does not induce intoxication or euphoria at typical doses and has a favourable safety profile compared to THC [12,13]. There is good evidence for efficacy of CBD in treating various types of treatment-resistant paediatric epilepsy [14,15] and modest evidence of benefits in treating anxiety and addictions [16,17]. With THC, there is low to moderate evidence of benefits in treating nausea and vomiting, muscle spasms, and cachexia [11,14]. There is also moderate evidence that THC can reduce non-cancer related pain including moderate to high quality evidence in favour of nabiximols (a combination THC and CBD in a 1:1 ratio) [14,18–20].

A recent review of TGA approval data reported that in 2020 approximately 74.8% of approvals were for Schedule 8 products (prescribed products containing >2% THC), and 25.6% were for Schedule 4 products (prescribed conducts containing >98% CBD) [9]. A better understanding of MC product composition is critical for evaluating the clinical significance and appropriateness of current MC prescribing and its effectiveness among prescribed medicinal cannabis users in Australia.

Since 2016, when MC was legalised in Australia, the most common indications for use have included pain, mental health, and sleep disorders [3,9]. Further research is required to understand how treatments of various indications vary by cannabinoid composition given that prescribed MC products with CBD and/or THC arguably have differential efficacy and safety, or lack thereof, in treating pain, mental health, and sleep [21]. While many health care providers, consumers and lay media describe MC as being one product–there are considerable differences in the pharmacological actions, clinical rationale, safety and effectiveness of THC and CBD products [21]. This paper will examine consumer experiences of using THC versus CBD products to better understand how these medicines are being used, and for what conditions they are being prescribed, in the Australian context.

The present study will address these research gaps through investigation of four aims: (1) to describe the demographic profile of prescribed MC consumers by cannabinoid product composition, (2) to examine how cost of, and patterns and reasons for, prescribed MC use differ by cannabinoid product composition and how this relates to the available clinical evidence, (3) to compare patient reported effectiveness and side-effect profiles of prescribed cannabinoid products, and (4) to review safety profiles of prescribed cannabinoid products in terms of concurrent medication use and driving patterns post-consumption

## Methods

We utilised data collected from an online anonymous cross-sectional survey of individuals (CAMS-20 survey) [3] who reported cannabis use for therapeutic or medical reasons within the preceding 12 months. Study data was collected and managed using Research Electronic Data Capture (REDCap), a secure web-based application [3,22]. To be included in this study, participants had to: i) self-identify using either prescribed MC only or both prescribed and illicit MC, and ii) be aware of the constituents of the prescribed MC (i.e. THC only vs THC-CBD vs CBD only). Any additional missing data was treated as missing where relevant in subgroup analyses, but this did not warrant exclusion of the participant from overall analysis. In total, 546 respondents were included in the study (N = 350 who reported using either both illicit and prescribed MC and N = 196 who reported using prescribed MC only). We

analysed their responses to survey questions examining: (1) demographic characteristics; (2) cannabinoid profile of *prescribed* product; (3) primary medical condition for use of *prescribed* cannabinoid; (4) self-reported side-effects; (5) self-reported improvement in symptoms and concurrent use of other medications in relation to the *prescribed* product over last 12 months; (6) self-reported cost of *prescribed* MC products and (7) self-reported time between any MC consumption and driving. Data from participants reporting use of illicit MC products only (N = 999) were excluded as these participants were very unlikely to know the cannabinoid composition of the products they were using. Data from participants receiving prescribed MC who were uncertain about its composition or who stated that it varied between batches (N = 19) were also excluded from analysis.

Complete details regarding the methodology for CAMS-20 are available elsewhere [3]. In summary, data were collected online through a secure web application, REDCap, from September 2020 to January 2021. Participants were invited to participate online using consumer group and social media webpages, at consumer and professional forums, as well as through several private medical cannabis clinics. To be eligible for inclusion in the study, participants had to provide informed consent and i) be aged 18 and above, ii) be an Australian resident and iii) have used cannabis or a cannabis-based product for a medical purpose within the last year.

Only participants who reported accessing and using prescribed MC were included in this analysis (n = 546). Prescribed MC users were defined as either i) participants who had regularly used prescribed MC within the past two weeks and intended to continue prescribed MC treatment (n = 473, 87%), ii) participants who used prescribed medicinal cannabis as and when required in the past 12 months and planned to continue prescribed MC treatment in the future (n = 28, 5%), and iii) participants who had used prescribed MC in the past 12 months but stopped more than two weeks ago with no intention to continue using (n = 45, 8%). We subgrouped these participants into three groups based on the composition of the prescribed cannabinoid product accessed:

- THC only (THC) group (n = 144) refers to participants who received prescribed cannabinoid products containing only THC or mostly (>98%) THC;

- THC-CBD (T+C) group (n = 227) refers to participants who received prescribed cannabinoid products containing significant amounts of both THC and CBD (>2% of both THC and CBD); and

- CBD only (CBD) group (n = 175) refers to participants who received prescribed cannabinoid products containing only CBD or mostly (>98%) CBD.

To examine route of administration, we grouped participants based on the main way they reported consuming prescribed medicinal cannabis. Participants who reported using tablets, edibles, liquid or spray were classified as 'oral'; those who reported using vaporisers were classified as 'vaporisers'; and those who reported using rolled-joints, pipes or "bongs" were classified as 'inhalers'. Only three participants were unable to be classified as such, (two who reported primarily using prescribed MC as a cream and one as a suppository), and were excluded from analysis.

To examine the side-effect profiles, we collapsed the THC and T+C groups into one group and labelled it the 'combined THC' group given the generally low doses of CBD in the cannabinoid products containing both THC and CBD. Notably, demonstrated efficacy of CBD tends to involve daily doses ≥300 mg, while therapeutic THC doses typically start around 10 mg [16,23]. Thus the significance of the CBD dose in respondents receiving a prescribed THC-CBD product with doses closer to 1:1 is very unclear.

We excluded all missing data from analyses. We utilised various simple linear regression models to analyse the data: Gaussian models were used for continuous outcomes such as age; binary logistic regression models for binary categorical outcomes such as relationship status and past use of illicit cannabis; multinomial logistic regression models for multilevel outcomes such as route of administration, and ordinal regression models for outcomes such as change in symptoms following MC treatment. The primary predictor of interest in these models was the covariate 'prescribed MC composition' which indicated whether the patient was in the THC, T+C or CBD group within the preceding 12 months. We utilised estimated marginal means to perform group contrasts within each regression model. We compared groups either utilising a three-way comparison involving all pairwise combinations of the three groups (i.e. THC vs T+C; THC vs CBD; T+C vs CBD) or a two-way comparison looking at the combined THC group versus the CBD group. We utilised the Bonferroni procedure to minimise the rate of falsely significant findings in the models involving three-way comparisons. We performed the above statistical analyses using STATA v18. Only p values <0.05 were considered statistically significant.

### Ethics approval and consent to participate

The CAMS-20 study was approved by the Sydney University Human Research Ethics Committee (2018/544). In order to begin the online survey, participants were asked to read a linked Participant Information Statement, and to tick a check box giving consent for their data to be used in the study.

## Results

Of the 2152 respondents who commenced the survey, 601 reported using prescribed medicinal cannabis in the preceding 12 months. Data was excluded for 55 participants who either did not indicate the composition of the prescribed MC (n = 36) or who were unsure or stated that the contents varied between batches (n = 19). Data are reported for the remaining 546 respondents of whom 80.2% of respondents completed all survey questions.

Most respondents who reported using prescribed MC products in the past 12 months reported using prescribed MC products containing a combination of THC and CBD (n = 227,41.6%) (T+C group); 144 (26.4%) reported using prescribed MC products containing only or mostly THC (THC group), and 175 (32.1%) reporting using prescribed MC products containing only or mostly CBD products (CBD group).

Basic demographic and between-group comparisons (Table 1) indicate that THC group participants were, on average, statistically younger than T+C ($M_{diff}$ = 3.8 (0.2,7.4), p<0.01) and CBD ($M_{diff}$ = 5.0 (1.2, 8.8), p = 0.03) group participants. Additionally, when compared with the CBD group, the THC group was more likely to be male (OR = 4.4(2.5, 7.8), p<0.001) and have regularly used illicit cannabis in the past (OR = 3.0 (1.8,5.0), p<0.001). There were no significant differences between the groups in terms of level of employment, education or relationship status.

Significant differences were also seen regarding cost of medication and route of administration (Table 1). Of those receiving prescribed medicinal cannabis only, the THC group spent on average $AU39.60 more per week on prescribed MC products than the CBD group (95% CI: $16.2-$63.0, p<0.001) and $AU45.0 more per week than the T+C group (95%CI:$20.8-$69.1, p<0.001). Additionally, the odds of the THC group using vaporised as opposed to oral prescribed MC products were greater than those of the T+C and CBD groups (OR = 4.8 (3.0,7.8), p<0.001; OR = 109.2(25.91,460.42), p<0.001 respectively).

**Table 1. Characteristics of prescribed medicinal cannabis users by cannabinoid product composition.**

| Characteristic | THC[a] (n = 144) | T+C[a] (n = 227) | CBD[a] (n = 175) | Total (n = 546) | Coefficient | Comparisons estimate (95% CI) | Standard p values | Bonferonni adjusted p values |
|---|---|---|---|---|---|---|---|---|
| **Age, numeric, in y, M (SD)** | 42.5 (13.2) | 46.3 (13.6) | 47.5 (14.9) | 45.6 (14.1) | Beta | **TO-T+C:-3.8 (-7.4,-0.2)** **TO-CO: -5.0 (-8.8,-1.2)** T+C–CO: -1.2 (-4.6, 2.2) | **0.002** **0.01** 0.39 | **0.005** **0.03** 1.00 |
| **Gender, binary[b] (not-male [ref] vs male), n (%) male** | 97 (67%) | 133 (59%) | 56 (32%) | 286 (52%) | Odds ratio | TO-T+C: 1.5 (0.9,2.5) **TO-CO:4.4 (2.5,7.8)** **T+C-CO:3.0 (1.8,5.0)** | 0.10 **<0.001** **<0.001** | 0.27 **<0.001** **<0.001** |
| **Highest level of education, binary, (school [ref] vs [tertiary] n (%) Tertiary** | 110 (76%) | 189 (84%) | 140 (80%) | 439 (80%) | Odds ratio | -TO-T+C: 0.7 (0.3,1.2) TO-CO:0.8 (0.4,1.6)T +C-CO:1.2 (0.7,2.3) | 0.10 0.40 0.44 | 0.31 1.00 1.00 |
| **Employment, binary[c], not-employed [ref] vs employed, n (%) employed** | 61 (42%) | 114 (50%) | 90 (51%) | 265 (49%) | Odds ratio | TO-T+C: 0.7 (0.4,1.2) TO-CO:0.7 (0.4,1.2) T+C-CO:1.0 (0.6,1.5) | 0.14 0.11 0.81 | 0.42 0.32 1.00 |
| **Relationship Status[d], binary, n (%), single[ref] vs partnered, n(%) partnered Partnered** | 83 (62%) | 151 (69%) | 104 (64%) | 338 (66%) | Odds ratio | TO-T+C: 0.7 (0.4,1.3) TO-CO:0.9 (0.5,1.6) T+C-CO:1.2 (0.7,2.1) | 0.09 0.75 0.14 | 0.53 1.00 0.99 |
| **Used non-medical cannabis regularly prior to commencing MC, binary, No [ref] vs Yes, n (%) Yes** | 118 (94%) | 194 (94%) | 135 (85%) | 447 (91%) | Odds ratio | TO-T+C: 1.1 (0.4,3.6) **TO-CO:3.0 (1.0,8.8)** **T+C-CO:2.7 (1.1–6.3)** | 0.80 **0.01** **<0.01** | 1.00 **0.04** **0.02** |
| **Weekly Cost of prescribed medicinal cannabis[e], numeric, in $AU, M (SD)** | 113.7 (78.4) | 68.8 (53.1) | 74.2 (49.2) | 79.6 (59.1) | Beta | **TO-T+C: 45.0 (20.8, 69.1)** **TO-CO: 39.6 (16.2, 63.0)** T+C-CO: -5.4 (-24.4, 13.6) | **<0.001** **<0.001** **0.58** | **<0.001** **<0.001** 0.58 |
| **Route of Administration, categorical[f], oral [ref], n (%) Vaporiser Inhaled** | 69 (50%) 16 (12%) | 45 (20%) 12 (5%) | 2 (1%) 0 (0%) | 116 (22%) 28 (5%) | Odds ratio | **TO-T+C: 4.8 (3.0, 7.8)** **TO-CO: 109.2 (25.91, 460.42)** **T+C-CO: 22.8 (5.4,95.3)** **TO-T+C: 4.2 (1.9, 9.4)** TO-CO: *Not enough data* T+C-CO: *Not enough data* | **<0.001** **<0.001** **<0.001** **<0.001** NA **NA** | **<0.001** **<0.001** **<0.001** **0.001** NA NA |
| **Frequency of use, numeric, days used in past 28 days, M (SD)** | 22.8 (9.2) | 23.3 (8.4) | 22.5 (9.4) | 22.9 (8.9) | Beta | TO-T+C: -0.5 (-2.9,1.9) TO-CO: 0.3 (-2.2, 2.9) T+C-CO: 0.8 (-1.5, 3.1) | 0.63 0.77 0.40 | 1.00 1.00 1.00 |

[a]THC = THC only, T+C = THC + CBD, CBD = CBD only

[b]Female (n = 254) and Other (n = 6) were collapsed into a single variable, not-male to facilitate analysis.

[c] Not employed included participants who were on the DSP, Home-duties, retired, or students. Employed included participants working full or part-time.

[d] Participants who selected 'Rather not say' were excluded from analysis. 'Single' was the reference category.

[e]Participants who indicated that they used both prescribed and illicit MC were excluded from analysis

[f]Two participants reported using CBD as a cream and one participant reported using THC as a suppository and were excluded from analysis.

## Patterns of prescribed medical cannabis use

The main conditions for which participants were prescribed MC are summarised in Table 2. Of the 498 respondents who indicated the condition they treated with prescribed MC, the majority were treating pain (n = 263, 53%), followed by mental health and substance use (n = 133, 27%); neurological symptoms (n = 34, 7%), sleep (n = 29, 6%), other (n = 22, 4%), gastrointestinal symptoms (n = 8, 2%) and cancer (n = 9, 2%). There was no statistically significant difference in the likelihood of participants selecting prescribed THC, T+C or CBD to manage the above umbrella presentations, however there were significant differences for specific conditions. Odds of the THC and T+C groups using prescribed MC to manage back pain

**Table 2. Main reasons for use of prescribed medicinal cannabis users by cannabinoid product production composition.**

| Indication | THC[a] (n = 144) | T+C [a] (n = 227) | CBD[a] (n = 175) | Comparisons estimate OR (95% CI)[r] | Standard p value | Bonferonni adjusted P value |
|---|---|---|---|---|---|---|
| **Pain (total), binary, No [ref] vs Yes, n(%) Yes (n = 263)** | 75 (29%) | 111 (42%) | 77 (29%) | TO-T+C: 1.1 (0.6,1.8) | 0.80 | 1.00 |
|  | 22 (37%) | 28 (47%) | 9 (15%) | TO-CO: 1.3 (0.7,2.3) | 0.24 | 0.73 |
| **Backpain, binary, No [ref] vs Yes, n(%) Yes (n = 59)** | 10 (19%) | 18 (35%) | 24 (46%) | T+C-CO: 1.2 (0.7,2.6) | 0.31 | 0.92 |
|  | 8 (20%) | 15 (38%) | 17 (43%) | TO-T+C: 1.3 (0.6,2.7) | 0.42 | 1.00 |
| **Arthritis, binary, No [ref] vs Yes, n(%) Yes (n = 52)** | 7 (21%) | 16 (47%) | 11 (32%) | **TO-CO: 3.3 (1.2,8.9)** | 0.004 | **0.01** |
|  | 11 (32%) | 15 (44%) | 8 (24%) | **T+C-CO: 2.6 (1.0,6.7)** | 0.02 | **0.049** |
| **Fibromyalgia, binary, No [ref] vs Yes, n(%) Yes (n = 40)** |  |  |  | TO-T+C: 0.9 (0.3,2.3) | 0.72 | 1.00 |
|  |  |  |  | TO-CO: 0.5 (0.2,1.2) | 0.06 | 0.17 |
|  |  |  |  | T+C-CO: 0.5 (0.2, 1.2) | 0.06 | 0.19 |
| **Neuralgia, binary, No [ref] vs Yes, n(%) Yes (n = 40)** |  |  |  | TO-T+C: 0.8 (0.3,2.5) | 0.68 | 1.00 |
|  |  |  |  | TO-CO: 0.5 (0.2,1.6) | 0.17 | 0.52 |
|  |  |  |  | T+C-CO: 0.7 (0.3, 1.6) | 0.26 | 0.77 |
| **CNCP[b], binary, No [ref] vs Yes, n(%) Yes (n = 34)** |  |  |  | TO-T+C: 0.6 (0.2,2.0) | 0.35 | 1.00 |
|  |  |  |  | TO-CO: 0.7 (0.2,2.4) | 0.53 | 1.00 |
|  |  |  |  | T+C-CO: 1.1 (0.4, 3.0) | 0.76 | 1.00 |
|  |  |  |  | TO-T+C: 1.2 (0.4,3.1) | 0.73 | 1.00 |
|  |  |  |  | TO-CO: 1.7 (0.5,5.4) | 0.33 | 0.76 |
|  |  |  |  | T+C-CO: 1.5 (0.5, 4.3) | 1.00 | 1.00 |
| **MHSU[c] (total), binary, No [ref] vs Yes, n(%) Yes (n = 133)** | 32 (24%) | 52 (39%) | 49 (37%) | TO-T+C: 0.9 (0.5,1.7) | 0.73 | 1.00 |
|  | 21 (25%) | 30 (34%) | 37 (42%) | TO-CO: 0.7 (0.4,1.3) | 0.16 | 0.49 |
| **Anxiety, binary, No [ref] vs Yes, n(%) Yes (n = 88)** | 4 (40%) | 6 (60%) | 2 (20%) | T+C-CO: 0.8 (0.4, 1.3) | 0.24 | 0.70 |
|  |  |  |  | TO-T+C: 1.1 (0.5,2.3) | 0.71 | 1.00 |
| **Depression, binary, No [ref] vs Yes, n(%) Yes (n = 10)** |  |  |  | TO-CO: 0.6 (0.3,1.3) | 0.13 | 0.40 |
|  |  |  |  | **T+C-CO: 0.6 (0.3, 1.1)** | **0.04** | 0.11 |
|  |  |  |  | TO -T+C: 1.1 (0.2,5.0) | 0.94 | 1.00 |
|  |  |  |  | TO-CO: 2.5 (0.3,20.0) | 0.30 | 0.90 |
|  |  |  |  | T+C-CO: 2.3 (0.3, 16.8) | 0.30 | 0.90 |
| **Neurological disorder, binary, No [ref] vs Yes, n(%) Yes (n = 34)** | 7 (21%) | 16 (47%) | 11 (32%) | TO -T+C: 0.6 (0.2,2.0) | 0.35 | 1.00 |
|  |  |  |  | TO-CO: 0.7 (0.2,2.4) | 0.53 | 1.00 |
|  |  |  |  | T+C-CO: 1.1 (0.4, 3.0) | 0.76 | 1.00 |
| **Sleep, binary, No [ref] vs Yes, n (%) Yes (n = 29)** | 11 (38%) | 9 (31%) | 9 (31%) | TO -T+C: 1.9 (0.6,5.9) | 0.16 | 0.47 |
|  |  |  |  | TO-CO: 1.5 (0.5,4.5) | 0.41 | 1.00 |
|  |  |  |  | T+C-CO: 0.8 (0.2, 2.4) | 0.57 | 1.00 |
| **Gastrointestinal disorder, binary, No [ref] vs Yes, n (%) Yes (n = 8)** | 1 (13%) | 5 (63%) | 2 (25%) | TO -T+C: 0.3 (0.02,4.2) | 0.27 | 0.82 |
|  |  |  |  | TO-CO: 0.6 (0.03,11.1) | 0.66 | 1.00 |
|  |  |  |  | T+C-CO: 2.0 (0.3, 14.7) | 0.43 | 1.00 |
| **Cancer, binary, No [ref] vs Yes, n (%) Yes (n = 9)** | 2 (22%) | 6 (67%) | 1 (11%) | TO -T+C: 0.5 (0.07,3.6) | 0.40 | 1.00 |
|  |  |  |  | TO-CO: 2.4 (0.1,44.9) | 0.49 | 1.00 |
|  |  |  |  | T+C-CO: 4.7 (0.4, 63.6) | 0.15 | 0.46 |
| **Other, binary, No [ref] vs Yes, n (%) Yes (n = 22)** | 7 (32%) | 7 (27%) | 9 (41%) | TO -T+C: 1.6 (0.4,5.9) | 0.39 | 1.00 |
|  |  |  |  | TO-CO: 0.9 (0.3,3.2) | 0.91 | 1.00 |
|  |  |  |  | T+C-CO: 0.6 (0.2, 2.0) | 0,30 | 0.90 |

[a] THC = THC only, T+C = THC + CBD, CBD = CBD only

[b] CNCP = Chronic Non Cancer Pain

[c] MHCSU = Mental Health and Substance Use.

were significantly greater than those of the CBD group (OR = 3.3 (1.2,8.9), p = 0.01 and OR = 2.6 (1.0,6.7), p = 0.05 respectively).

## Side effects

The most commonly reported side effects in those receiving prescribed MC were: dry mouth (n = 261, 56%); drowsiness (n = 239, 51%); fatigue (n = 129, 28%); eye irritation (n = 113, 24%), anxiety (n = 87, 19%), dizziness (n = 82, 18%); bad taste (n = 71, 15%) and confusion (n = 59, 13%). Odds were statistically greater of the THC and T+C groups reporting dry

**Table 3. Most commonly reported side effects in participants receiving prescribed MC.**

| Side Effect | THC[a] (n = 313) | CBD[b] (n = 152) | Total (n = 465) | Coefficient | Comparisons estimate (95% CI) | p-values |
|---|---|---|---|---|---|---|
| **Dry Mouth,** No [ref] vs Yes, n(%) | 197 (63%) | 64 (42%) | 261 (56%) | Odds Ratio | **THC-CO: 2.3 (1.6, 3.5)** | **<0.001** |
| **Drowsiness,** No [ref] vs Yes, n(%) | 179 (57%) | 60 (39%) | 239 (51%) | Odds Ratio | **THC-CO: 2.0 (1.4,3.0)** | **<0.001** |
| **Fatigue,** No [ref] vs Yes, n(%) | 88 (28%) | 41 (27%) | 129 (28%) | Odds Ratio | THC-CO: 1.1 (0.7, 1.6) | 0.80 |
| **Eye Irritation,** No [ref] vs Yes, n(%) | 92 (29%) | 21 (14%) | 113 (24%) | Odds Ratio | **THC-CO: 2.6 (1.5, 4.4)** | **<0.001** |
| **Anxiety,** No [ref] vs Yes, n(%) | 55 (18%) | 32 (21%) | 87 (19%) | Odds Ratio | THC-CO: 0.8 (0.5, 1.3) | 0.37 |
| **Dizziness,** No [ref] vs Yes, n(%) | 61 (19%) | 21 (14%) | 82 (18%) | Odds Ratio | THC-CO: 1.5 (0.9, 2.6) | 0.14 |
| **Bad Taste,** No [ref] vs Yes, n(%) | 47 (15%) | 24 (15%) | 71 (15%) | Odds Ratio | THC-CO: 0.9 (0.6, 1.6) | 0.83 |
| **Confusion,** No [ref] vs Yes, n(%) | 42 (13%) | 17 (11%) | 59 (13%) | Odds Ratio | THC-CO: 1.2 (0.7, 2.2) | 0.50 |

mouth (OR = 2.3 (1.6,3.5), p<0.001), drowsiness (OR = 2.0 (1.4,3.0), p<0.001) and eye-irritation (OR = 2.6 (1.5,4.4)) than the CBD group (Table 3).

## Effectiveness of prescribed medical cannabis

Participants utilised the Patients' Global Impression of Change (PGIC) [24] measuring tool to indicate the effect of prescribed MC on the main condition treated in the past 12 months. The PGIC consists of seven items: "Very much better"; "Much better"; "A little better"; "No change"; "A little worse"; "Much worse" and "Very much worse". Patients were asked to "rate how their <main condition> has changed since using prescribed medicinal cannabis" The overwhelming majority of participants receiving prescribed MC reported improvement (recorded by PGIC as "a little better", "much better" or "very much better") in pain (n = 209, 95%), depression (n = 10, 100%), anxiety (n = 85, 97%) and sleep (n = 30, 83%).

Odds were significantly greater for participants receiving prescribed products containing THC reporting any improvements in pain (OR = 2.2(1.1,4.7), p = 0.03), mental health and substance use (OR = 3.1(1.1,9.0), p = 0.03) and anxiety (OR = 4.5(1.4,14.9), p<0.007) than those prescribed CBD-dominant products. No statistically significant differences in patient-rated effectiveness of THC-dominant products or T+C products as opposed to CBD-dominant products were noted for any of the other indications (See Table 4).

## Concurrent use of other medications

Most participants receiving prescribed MC reported using other prescribed medications including: opioids (n = 310, 56%); benzodiazepines (n = 266, 50%); antidepressants (n = 287, 53%); antipsychotics (n = 71; 13%); anticonvulsants (n = 50;9%); gabapentinoids (n = 117, 21%) and non-opioid analgesics (NSAIDs, paracetamol) (n = 285, 52%). Odds of participants on prescribed THC-CBD also being prescribed opioids were significantly greater than that of those on prescribed CBD only (OR = 1.6(1.0,2.6), p = 0.05), and odds of participants on prescribed THC being prescribed antipsychotics were greater than those on either prescribed THC-CBD or CBD only (OR = 2.1(1.0,4.3), p = 0.03 and OR = 2.6(1.2,5.8), p = 0.01 respectively). No other significant differences in rates of concurrent medication use were noted between the three groups (Table 5).

We performed analyses related to driving on the subgroup of participants who took prescribed MC only, not those who took both prescribed and illicit MC. Most of these participants reported having driven in the last year (n = 364, 89%). Moreover, there was no significant difference in odds of consumers of prescribed MC containing THC driving within twenty-four, twelve, or six hours of MC consumption than consumers of prescribed MC containing CBD only, despite current research advising that impairment associated with THC lasts about 6 hours from inhaled use

**Table 4. Reported improvement in main condition: comparing prescribed cannabinoids using PGIC scores.**

| Degree of Improvement in main condition | THC (n = 144)[a] | T+C (n = 227)[a] | CBD (n = 175)[a] | Odds of a greater degree of improvement | Standard p values | Bonferonni adjusted p values |
|---|---|---|---|---|---|---|
| **Pain,** numeric, M (SD) | 6.1 (0.8) | 6.1 (0.9) | 5.8 (0.9) | TO-T+C: 0.6 (0.3, 1.3)<br>TO-CO: 1.3 (0.6,2.9)<br>**T+C-CO: 2.2 (1.1,4.7)** | 0.10<br>0.40<br>**0.01** | 0.28<br>1.00<br>**0.03** |
| **MHSU[b]** numeric, M (SD) | 6.4 (0.7) | 6.2 (1.0) | 5.9 (1.0) | TO-T+C: 1.5 (0.5,4.1)<br>**TO-CO: 3.1 (1.1,9.0)**<br>T+C-CO: 2.1 (0.9,5.3) | 0.38<br>**0.01**<br>**0.05** | 1.00<br>**0.03**<br>0.14 |
| **Anxiety,** numeric, M (SD) | 6.3 (0.7) | 6.5 (0.6) | 5.9 (0.9) | TO-T+C: 0.6 (0.2,2.3)<br>TO-CO: 2.8 (0.8,10.3)<br>**T+C-CO: 4.5 (1.4,14.9)** | 0.40<br>**0.05**<br>**0.002** | 1.00<br>0.15<br>**0.007** |
| **Depression,** numeric, M (SD) | 5.8 (0.5) | 6.2 (0.8) | 6 (0) | TO-T+C: 0.2 (0.0,7.5)<br>TO-CO: 0.5 (0.0,71.8)<br>T+C-CO: 2.2 (0.0, 339.4) | 0.30<br>0.71<br>0.71 | 0.89<br>1.00<br>1.00 |
| **Neuro,** numeric, M (SD) | 6.1 (0.7) | 6.1 (0.9) | 5.8 (0.8) | TO-T+C: 0.9 (0.1,14.0)<br>TO-CO: 2.1 (0.3,17.5)<br>T+C-CO: 2.3 (0.4,14.0) | 0.91<br>0.40<br>0.26 | 1.00<br>1.00<br>0.79 |
| **Sleep,** numeric, M (SD) | 6.3 (0.7) | 6.7 (0.5) | 5.3 (1.4) | TO-T+C: 0.9 (0.1,8.2)<br>TO-CO: 5.6 (0.6,48.1)<br>T+C-CO: 6.3 (0.6,61.9) | 0.90<br>0.06<br>0.06 | 1.00<br>0.17<br>0.17 |
| **Gastro** numeric, M (SD) | 5 (NA) | 7 (0) | 6.5 (0.7) | Insufficient data | | Insufficient data |
| **Cancer,** numeric, M (SD) | 5.5 (0.7) | 5.7 (1.5) | 5 (NA) | TO-T+C: 0.5 (0.0,14.6)<br>TO-CO: 2.6 (0.0,314.8)<br>T+C-CO: 5.4 (0.1,489.4) | 0.60<br>0.64<br>0.37 | 1.00<br>1.00<br>1.00 |
| **Other,** numeric, M (SD) | 6.6 (0.5) | 5.8 (0.8) | 6.3 (0.7) | TO-T+C: 8.7 (0.5,149.1)<br>TO-CO: 1.9 (0.2, 20.3)<br>T+C-CO: 0.2 (0.0,3.1) | 0.07<br>0.51<br>0.17 | 0.20<br>1.00<br>0.50 |
| **Back pain,** numeric, M (SD) | 6.1 (0.6) | 6.0 (0.8) | 5.6 (1.1) | TO-T+C: 1.2 (0.3,4.6)<br>TO-CO: 2.9 (0.5, 17.9)<br>T+C-CO: 2.3 (0.4,14.0) | 0.68<br>0.16<br>0.26 | 1.00<br>0.49<br>0.79 |
| **Arthritis,** numeric, M (SD) | 5.8 (1.1) | 6.2 (0.7) | 6.2 (0.7) | TO-T+C: 0.5 (0.1,3.6)<br>TO-CO: 0.5 (0.1, 3.3)<br>T+C-CO: 1.0 (0.3,4.0) | 0.44<br>0.42<br>1.00 | 1.00<br>1.00<br>1.00 |
| **Fibromyalgia,** numeric, M (SD) | 5.9 (0.6) | 6.3 (0.7) | 5.9 (0.8) | TO-T+C: 0.3 (0.0,2.1)<br>TO-CO: 0.8 (0.1, 5.1)<br>T+C-CO: 2.7 (0.5,14.6) | 0.13<br>0.74<br>0.15 | 0.39<br>1.00<br>0.45 |
| **Nerve Pain,** numeric, M (SD) | 6.3 (0.8) | 6.0 (1.0) | 6.1 (0.8) | TO-T+C: 1.5 (0.3,7.4)<br>TO-CO: 1.4 (0.2, 9.4)<br>T+C-CO: 0.9 (0.1,6.5) | 0.55<br>0.70<br>0.91 | 1.00<br>1.00<br>1.00 |
| **CNCP[c] (undefined),** numeric, M (SD) | 6 (0.8) | 6.1 (1.0) | 5.5 (1.1) | TO-T+C: 0.6 (0.1,3.6)<br>TO-CO: 1.9 (0.3, 14.1)<br>T+C-CO: 3.3 (0.5,23.2) | 0.49<br>0.42<br>0.15 | 1.00<br>1.00<br>0.44 |

[a] THC = THC only, T+C = THC + CBD, CBD = CBD only

[b] CNCP = Chronic Non Cancer Pain

[c] MHSU = Mental Health and Substance Use.

whilst minimal impairment is associated with CBD [25]. However, the odds of consumers of prescribed MC containing THC driving within three and one hour of consumption were significantly lower than that of consumers of prescribed MC containing CBD only [25] (OR = 0.6 (0.4, 1.0), p = 0.04, and OR = 0.4 (0.2,0.6), p<0.001 respectively) (Table 6).

## Discussion

This study is one of the first to examine the characteristics of users of different formulations of prescribed MC in Australia. Possession of prescribed MC has been legal in Australia since

**Table 5. Other medication used versus prescribed cannabinoids.**

| Other medication use | THC[a] (n = 144) | T+C[a] (n = 227) | CBD[a] (n = 175) | Total (n = 546) | Comparisons estimate (95% CI)[r] | P value | Bonferonni adjusted p value |
|---|---|---|---|---|---|---|---|
| **Opioids,** binary, No [ref] vs Yes, n(%) | 78 (54%) | 142 (63%) | 90 (51%) | 310 (56%) | TO-T+C: 0.7 (0.4,1.2)<br>TO-CO: 1.1 (0.7,1.9)<br>**T+C-CO: 1.6 (1.0,2.6)** | 0.11<br>0.63<br>**0.03** | 0.33<br>1.00<br>**0.05** |
| **Benzodiazepines,** binary, No [ref] vs Yes, n(%) | 74 (51%) | 111 (49%) | 81 (46%) | 266 (50%) | TO-T+C: 1.1 (0.7,1.8)<br>TO-CO: 1.2 (0.7,2.1)<br>T+C-CO: 1.1 (0.7,1.8) | 0.60<br>0.36<br>0.64 | 1.00<br>1.00<br>1.00 |
| **Antidepressants,** binary, No [ref] vs Yes, n(%) | 80 (56%) | 114 (50%) | 93 (53%) | 287 (53%) | TO-T+C: 1.2 (0.7,2.1)<br>TO-CO: 1.1 (0.6, 1.9)<br>T+C-CO: 0.9 (0.5,1.4) | 0.32<br>0.67<br>0.56 | 0.95<br>1.00<br>1.00 |
| **Antipsychotics,** binary, No [ref] vs Yes, n(%) | 30 (21%) | 25 (11%) | 16 (9%) | 71 (13%) | **TO-T+C: 2.1 (1.0,4.3)**<br>**TO-CO: 2.6 (1.2,5.8)**<br>T+C-CO: 1.2 (0.5, 2.8) | **0.01**<br>**0.004**<br>0.54 | **0.03**<br>**0.01**<br>1.00 |
| **Anticonvulsants,** binary, No [ref] vs Yes, n(%) | 16 (11%) | 17 (7%) | 17 (10%) | 50 (9%) | TO-T+C: 1.5 (0.6, 3.7)<br>TO-CO: 1.2 (0.5,2.8)<br>T+C-CO: 0.8 (0.3,1.8) | 0.24<br>0.68<br>0.43 | 0.71<br>1.00<br>1.00 |
| **Gabapentinoids,** binary, No [ref] vs Yes, n(%) | 33 (23%) | 50 (22%) | 34 (19%) | 117 (21%) | TO-T+C: 1.1 (0.6,1.9)<br>TO-CO: 1.2 (0.6,2.4)<br>T+C-CO: 1.2 (0.6,2.1) | 0.84<br>0.45<br>0.53 | 1.00<br>1.00<br>1.00 |
| **Non-opioid analgesics,** binary, No [ref] vs Yes, n(%) | 72 (50%) | 126 (56%) | 88 (50%) | 285 (52%) | TO-T+C: 0.8 (0.5, 1.3)<br>TO-CO: 1.0 (0.6,1.7)<br>T+C-CO: 1.2 (0.8,2.0) | 0.30<br>0.96<br>0.30 | 0.90<br>1.00<br>0.90 |
| **Other,** binary, No [ref] vs Yes, n(%) | 11 (8%) | 10 (4%) | 12 (7%) | 33 (6%) | TO-T+C: 1.8 (0.6,5.3)<br>TO-CO: 1.1 (0.4,3.2)<br>T+C-CO: 0.6 (0.2,1.8) | 0.19<br>0.79<br>0.29 | 0.58<br>1.00<br>0.86 |

[a] TO = THC only, T+C = THC + CBD, CO = CBD only.

2016, whilst possession of illicit cannabis, regardless of the purpose it serves, remains illegal with punishments varying from state to state [3]. Individuals primarily using prescribed THC-dominant products (TO group) were statistically younger, more likely to be male, and more likely to consume their prescribed product via inhalation compared to their CBD-dominant (CBD group) counterparts. They also spent on average $AU39.60 more on prescribed cannabinoids products per week but showed no significant difference in terms of frequency of use. Other demographics such as relationship status, employment and education did not differ between the groups, suggesting they do not play a significant role in terms of client preference.

**Table 6. Driving patterns of participants on prescribed THC vs prescribed CO.**

| Driving activities | THC[a] (n = 241) | CO[a] (n = 123) | Total (n = 364) | Coefficient | Comparisons estimate (95% CI) | p value |
|---|---|---|---|---|---|---|
| **Driving within 24 hours of consumption,** No [ref] vs Yes, n(%) | 210 (87%) | 113 (92%) | 323 (89%) | Odds Ratio | 0.6 (0.3,1.3) | 0.18 |
| **Driving within 12 hours of consumption,** No [ref] vs Yes, n(%) | 180 (75%) | 98 (80%) | 278 (76%) | Odds Ratio | 0.8 (0.4,1.3) | 0.29 |
| **Driving within 6 hours of consumption,** No [ref] vs Yes, n(%) | 117 (49%) | 69 (56%) | 186 (51%) | Odds Ratio | 0.7 (0.5, 1.2) | 0.17 |
| **Driving within 3 hours of consumption,** No [ref] vs Yes, n(%) | 85 (35%) | 57 (46%) | 142 (39%) | Odds Ratio | 0.6 (0.4,1.0) | **0.04** |
| **Driving within 1 hours of consumption,** No [ref] vs Yes, n(%) | 36 (15%) | 40 (33%) | 76 (21%) | Odds Ratio | 0.4 (0.2,0.6) | **<0.001** |

[a] THC = THC only or T+C, CBD = CBD only.

Systematic reviews have concluded that cannabinoids show limited efficacy as a monotherapy in reducing non-cancer pain [26–33], treating mental illnesses [34–38], and treating epilepsy [33,39,40] (with the exception of treating drug-resistant epilepsy for which there is strong evidence [14,15]), with modest improvements at best. Other systematic reviews conducted by the TGA have found that currently there is some evidence of efficacy for both THC and CBD in treatment of chronic non-cancer related pain [18] and neuropathic pain [41] (although both tend to favour THC-containing products). This was supported by our study in that prescribers were more likely to prescribe MC products containing THC to manage back pain [41]. Further systematic reviews also showed that CBD used as an adjunct to antiepileptic medications may reduce seizure frequency and improve quality of life in children under 25 years [15]. Whilst a recent RCT conducted in Australia demonstrated findings favouring cannabinoids (THC+CBD) in the treatment of chemotherapy induced nausea and vomiting [42], evidence for the role of cannabinoids in the treatment of mental health conditions such as anxiety [34–38] and in palliative care [42] is limited, and, as with all unlicensed medications, the current recommendation is for patients to trial other treatments first. This means that although prescribed MC is currently being used to treat a variety of different clinical syndromes with SAS-B approval, it is often with little definitive evidence [43]. Nonetheless over 46% of participants in this study received prescribed MC products for conditions other than pain and epilepsy demonstrating the variety of conditions for which patients seek cannabinoids and the ease at which doctors prescribe unregistered cannabis-based medicines to their patients regardless of their condition [21].

MacPhail et al (2022) examined trends in prescription patterns by indications under SAS-B and found that, since 2018, there has continued to be an exponential growth in approvals for Schedule 8 (S8) products whilst the growth in approvals for Schedule 4 (S4) products appears to be plateauing [9]. Around January 2021 it was estimated that approximately 67.0% of approvals were for S8 products which is very similar to the CAMS-20 data estimates of 68.0% recorded around the same time.

Additionally, the S4 versus S8 prescription patterns by indications under SAS-B reported by

Macphail et al are, for the most part, comparable to those seen in the CAMS-20 data, supporting the reliability of the CAMS-20 dataset (Table 7).

Unlike CBD, THC is well known for its increased risk of impairment [44,45], particularly when combined with other sedatives [20]. Nonetheless, our analyses revealed that patients on prescribed opioids and/or antipsychotics, drugs with sedating properties, are significantly more likely to be receiving THC-dominant or T+C products (70%) than CBD-dominant (30%) products, OR = 2.2 (1.0,4.6); perhaps suggesting a failure to investigate safer alternatives or inadequate attention to potential drug-drug interactions. Conversely, the primary concern with CBD is its risk of interaction with CYP450 metabolised drugs, particularly those engaging CYP3A4, CYP2C9 and CYP 2C19 [20,21,44,46,47]. CYP3A4 enzymes [21] are responsible for

**Table 7. MacPhail vs CAMS approximate (%) S8 approval in Jan 2021 for most common indications.**

| Indication | MacPhail S8 approval (%) | CAMS S8 approval (%) |
|---|---|---|
| Pain | 75 | 71 |
| Anxiety | 65 | 58 |
| Sleep Disorder | 85 | 69 |
| Cancer and Related Symptoms | 85 | 89 |
| Neuropathy | 75 | 68 |
| Post-Traumatic Stress Disorder | 75 | 74 |

the metabolism of benzodiazepines including alprazolam, midazolam and clobazam; and for part of the metabolism of other benzodiazepines including diazepam and flutrinazepam [48]. Thus, using CBD in conjunction with these benzodiazepines could result in enhanced sedation and possible toxicity. It is of some concern then that almost 50% of patients on prescribed CBD were also receiving concurrent benzodiazepines, although respondents who disclosed benzodiazepine use were less likely to be using CBD (30%) than THC products (70%) (odds ratio not statistically significant). Finally, there was no difference in the likelihood that participants consuming prescribed THC and participants consuming prescribed CBD would drive within 6, 12 and 24 hours of prescribed MC consumption. This is despite the fact that current research advises that impairment associated with THC lasts about 6 hours from inhaled use whilst minimal impairment is associated with CBD. Moreover, it is illegal to drive in Australia with any presence of THC in oral fluid, blood, or urine [49], suggesting that this should be more rigorously considered by both prescribing practitioners and patients.

Although multiple studies including systematic reviews, randomised trials and case series [26–34,43] have examined prescribed MC use in its various forms and have had favourable results, methodological barriers across studies, such as ineffective blinding of participants, exclusion of patients with comorbidities, reliance on participants self-reporting their cannabis exposure, use of other treatments to manage their conditions, and short study intervals (for chronic conditions) has meant the evidence produced by these studies is often weak at best.

There are limitations to this study. First, all data in the CAMS-20 survey is self-reported, meaning that there may be inaccuracies around diagnostic conditions, reported effectiveness and reported adverse events. Also, the fact that it was collected via an anonymous online survey meant that the sample may not be representative of prescribed MC users generally, with people with more favourable experiences perhaps more likely to participate due to conformation bias [3]. Finally, our decision to combine the THC and T+C groups in some analyses on the basis that the CBD effects would likely be minimal in groups receiving doses closer to 1:1; did not take into consideration the THC-CBD users who are predominantly receiving high CBD and low THC doses (e.g. 400mg CBD vs 20mg THC); however compositions such as this are rare.

## Conclusion

In Australia, consumers of prescribed MC products primarily containing THC tend to be younger, have higher rates of inhaled use, and are more likely to have used illicit cannabis in the past than those consuming prescribed MC products containing CBD only. Results from this study show a favouring of THC products over CBD for management of pain, mental health, anxiety, and sleep, with high rates of patient-reported improvement in conditions treated, however more rigorous investigations are crucial. Educating medical practitioners prescribing cannabinoids is essential to ensure cannabinoids are prescribed for appropriate reasons and to prevent avoidable drug interactions.

## Acknowledgments

We would like to acknowledge the important contribution of the late Associate Professor David Allsop in the creation of the original CAMS-16 questionnaire, upon which the CAMS-20 questionnaire was based. We would also like to acknowledge the contributions of Professor Jonathon Arnold, Dr Melissa Benson, Dr Dilara Bahceci and Rhys Cohen in this work, and the services and forums that distributed information regarding the survey. We would also like to thank the participants who gave their time in completing the survey.

## Author Contributions

**Conceptualization:** Benjamin T. Trevitt, Sasha Bailey, Llewellyn Mills, Thomas R. Arkell, Anastasia Suraev, Iain S. McGregor, Nicholas Lintzeris.

**Data curation:** Llewellyn Mills, Thomas R. Arkell, Anastasia Suraev, Iain S. McGregor, Nicholas Lintzeris.

**Formal analysis:** Benjamin T. Trevitt, Sasha Bailey.

**Investigation:** Benjamin T. Trevitt, Sasha Bailey, Llewellyn Mills, Thomas R. Arkell, Anastasia Suraev, Iain S. McGregor, Nicholas Lintzeris.

**Methodology:** Benjamin T. Trevitt, Sasha Bailey, Llewellyn Mills, Thomas R. Arkell, Anastasia Suraev, Iain S. McGregor, Nicholas Lintzeris.

**Resources:** Benjamin T. Trevitt, Sasha Bailey.

**Supervision:** Nicholas Lintzeris.

**Writing – original draft:** Benjamin T. Trevitt, Sasha Bailey.

**Writing – review & editing:** Benjamin T. Trevitt, Sasha Bailey, Llewellyn Mills, Thomas R. Arkell, Anastasia Suraev, Iain S. McGregor, Nicholas Lintzeris.

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
