## [Decision Letter · Decision Letter 0]

1 Sep 2023

PONE-D-23-23895Differences in prescribed medicinal cannabis use by cannabinoid product composition: Findings from the CAMS-20 Australia-wide studyPLOS ONE

Dear Dr. Trevitt,

Thank you for submitting your manuscript to PLOS ONE. After careful consideration, we feel that it has merit but does not fully meet PLOS ONE’s publication criteria as it currently stands. Therefore, we invite you to submit a revised version of the manuscript that addresses the points raised during the review process.

We look forward to receiving your revised manuscript.

Kind regards,

Andrea Mastinu

Academic Editor

PLOS ONE

3. We note that you have a patent relating to material pertinent to this article. Please provide an amended statement of Competing Interests to declare this patent (with details including name and number), along with any other relevant declarations relating to employment, consultancy, patents, products in development or modified products etc. Please confirm that this does not alter your adherence to all PLOS ONE policies on sharing data and materials, as detailed online in our guide for authors http://journals.plos.org/plosone/s/competing-interests by including the following statement: "This does not alter our adherence to  PLOS ONE policies on sharing data and materials.” If there are restrictions on sharing of data and/or materials, please state these. Please note that we cannot proceed with consideration of your article until this information has been declared.

Reviewers' comments:

Reviewer's Responses to Questions

**Comments to the Author**

1. Is the manuscript technically sound, and do the data support the conclusions?

Reviewer #1: Yes

Reviewer #2: Yes

Reviewer #3: Yes

Reviewer #4: Partly

Reviewer #5: Yes

Reviewer #6: Partly

2. Has the statistical analysis been performed appropriately and rigorously? 

Reviewer #1: I Don't Know

Reviewer #2: Yes

Reviewer #3: Yes

Reviewer #4: Yes

Reviewer #5: I Don't Know

Reviewer #6: Yes

3. Have the authors made all data underlying the findings in their manuscript fully available?

Reviewer #1: No

Reviewer #2: Yes

Reviewer #3: Yes

Reviewer #4: Yes

Reviewer #5: Yes

Reviewer #6: No

4. Is the manuscript presented in an intelligible fashion and written in standard English?

Reviewer #1: Yes

Reviewer #2: Yes

Reviewer #3: Yes

Reviewer #4: Yes

Reviewer #5: Yes

Reviewer #6: Yes

5. Review Comments to the Author

Reviewer #1: In this manuscript the authors have analysed the data they collected from their CAMS-20 survey of patients on their user experience in Australia who were taking a medicinal product (medicine) that contained the active ingredients of cannabidiol (CBD) and/or tetrahydrocannabinol (THC).

This work is a follow on from analysis already conducted and published on the CAMA-20 study; reference 6 in this manuscript.

I recommend acceptance pending the following points:

1. Lines 82 to 84. This statement about evidence for cannabis for non-cancer related pain cites TGA documents rather than research papers. I would recommend two recently completed Australian studies that looked at efficacy and safety of cannabis for pain as better references (British Journal of Pain, 2023, 17, 2, 206-217 and International Journal of Pharmacy Practice, 2023, 31, 70-79). The second reference is relevant for the authors discussion of co-medications in their manuscript (Lines 239-242 and 316-330).

2. In the introduction is a bit circle and repeats itself. For example lines 61-63 and 93-94 cover similar information. The introduction should be revised to make it more streamlined.

3. It's not clear what the inclusion criteria for the subgroup of the 546 participants was (line 117). I'm guessing that it was any participant that took a legal form of cannabis (implied by line 137) and participants with missing data (line 151) but that is ambiguous and should be made more clear at line 117.

4. Line 151. What constituted "missing data". How much data had to be "missing" before it was excluded?

5. Details of the statistic tests undertaken, and not just the software, need to be given (line 163).

6. The research team put products into three catgeories TO, T+C, and CO, but the TGA defines five categories of products. Would it not have been better to evaluate the products based on the TGA categories? Particularly when later (lines 180/181) the terms "mostly THC" and "mostly CBD" products are used.

7. Not sure I agreed with the statement (line 183) "TO group participants were significantly younger than T+C and CO group participants" The average age of TO was 42.5 and average age of the others if 46.3 and 47.5. Those are very similar ages. I think the authors mean there was a statistical difference in their average ages, but not that they were significantly younger. If this is changed here, then it also needs to be changed in the abstract and in the discussion (Line 265).

8. On what basis do the authors make the following statement about employment status (lines 184 to 186) "Additionally, when compared with CO users, TO users were more likely to be male, unemployed, and have regularly used illicit cannabis in the past" given their table finds no statistical difference between any of the three categories.

Reviewer #2: The article by Trevitt et al decribes the results of a survey of medicinal cannabis users, this is a well written article highlighting that the major benefits of medicinal cannabis come from predominately THC containing products and not CBD products.

Reviewer #3: The submission by Mr. Benjamin Trevitt and colleagues presents findings from a cross-sectional study of prescribed medical cannabis users in Australia that sought to better understand use of different product compositions and formulations, cost of product use, and self-reported change in symptoms associated with the primary medical indication for use over time. While several published cross-sectional studies have examined similar questions, the novelty of the current study is its focus on medical cannabis use in Australia, a county where medical cannabis is legal and adult-use cannabis (recreational) is illegal. Participants were assigned to one of three groups based on their provided medical cannabis formulation (THC, THC + CBD, and CBD). Converging with studies published in other countries, THC products were endorsed more frequently among young male users with inhalation as the preferred route of administration. Compared to participants using CBD products, participants spent more on THC products and these same individuals self-reported improvements in the conditions/symptoms for which they were prescribed medical cannabis. The manuscript is concise, organized, and well-written. Below, please find my comments and feedback addressed to the attention of the authors.

Major Concern(s):

Comment: N/A

Minor Concern(s):

Title

Comment: The authors should consider spelling out the full name of the survey study in the title before using an acronym (CAMS-20).

Abstract

Comment: For clarity, it would be helpful if the authors specified that participants are Australian for both the Methods and Results sections of the abstract. The authors should also consider including the study’s country of origin in the Conclusions section of the abstract, which sets this study apart from existing cross-sectional work published in other countries.

Background

Comment (General): The focus on Australia is important because it builds a knowledge base that other countries can refer to when considering reforms to cannabis legislation. Thus, it may be important to describe the current legal status of medical and non-medical cannabis in Australia for additional context.

Comment (Page 4, Lines 55-56): Please describe what is meant by “…pro-MC scheduling recommendation..”.

Comment (Page 4, Line 57): Since the location (Australia) is the setting for the study, readers in other countries need a few more details that explains the purpose of the Therapeutic Goods Administration (TGA) and Special Access Scheme-B (SAS-B).

Comment (Page 5, Lines 86-91): To distinguish the current work from existing findings, the authors should consider specifying that a “…better understanding of MC product composition is critical…’ among prescribed medical cannabis users in Australia.

Comment (Pages 5-6, Lines 94-105): Importantly, the author’s state how it’s important to understand how medical cannabis is being used in the Australian context. Study location aside, a number of cross-sectional retrospective and prospective survey studies have been published in other countries, which raises the question of the value of the current work. It may be useful to tighten up the rationale/argument for conducting the current study earlier in the Background.

Comment (Page 6, Lines 107-112): The authors present the results of analyses that examined concurrent medication use, but this is not listed in any of the study’s three aims (but discussed in the context of survey questionnaire content).

Methods

Comment (Page 6, Lines 117-119): Were the authors able to verify legal prescribed medical cannabis use or was it based on participant self-report?

Comment (Page 7, Lines 130-135): Please describe how study participants were recruited. Was any financial remuneration given to participants? The authors refer the reader to an existing publication, but these two details can be described concisely here.

Comment (Pages 7-8, Lines 140-145): The current abbreviations for each of the three groups (TO, T+C, CO) could be improved for greater clarity. For example, could these be changed to THC, THC + CBD, CBD, or something similar? The current abbreviations are not that intuitive.

In this same section, it would be helpful to specify the sample size for each of the three groups.

Comment (Page 8, Line 151): Were any methods to handle missing data considered?

Comment (Page 8, Lines 162-162): The data presented in each Table seems to indicate that comparisons were made using more than 50 separate variables. The authors used Bonferroni procedure to correct for multiple comparisons. It would be helpful to include a Bonferroni-adjusted p-value column in addition to the standard p-value column for the sake of data interpretation.

Comment (Page 8, Lines 166-167): How did participants complete the survey? Computer, tablet, mobile phone, all of the above?

Results

Comment (Page 9, Line 172): Please indicate the total number of participants that were screened for the study and provide the survey completion rate.

Comment (Pages 9-10, Lines 183-94): Please provide the results of the statistical tests and p-values when discussing the results in the main text.

Comment (Page 10, Lines 191-192): Please specify the currency that corresponds to the amount participants spend on medical cannabis products.

Comment (Page 13, Lines 202-210): Please provide the results of statistical tests and p-values for the comparisons discussed here.

Comment (Page 13, Line 204): For patient-reported conditions involving mental health and substance use, are the authors able to extract any specific mental health conditions (anxiety, depression, PTSD) in the current sample, or was mental health and substance use a broad category.

Comment (Page 16, Lines 214-218): The authors might consider ordering the side effects listed in Table 3 by their reported prevalence in the study (for example, from highest to lowest). Please also report the p-values for this section.

Comment (Page 16, Lines 223-227): It would be helpful to briefly detail the psychometric properties of the PGIC and the complete wording of the question that required participants to rate symptom improvement. Also, what is the time window that participants are using to report on their symptoms?

Comment (Page 18, Lines 239-246): Please include relevant test statistics and p-values.

Comment (Page 19, Lines 250-257): Please include relevant test statistics and p-values.

Discussion

Comment (Page 19, Lines 263-264): It may be useful to briefly state the current legal status of medical and non-medical cannabis in Australia.

Comment (Pages 20-21, Lines 272-299): The authors discuss the methodological barriers of previous studies but don’t discuss here whether the current study had any major limitations.

Comment (Page 22, Line 314): The location of Table 7 should be moved out of the discussion section.

Comment (Page 23, Lines 346-353): The concluding paragraph does include any instance of the word “Australia,” which is the major element of how this study differs from existing work.

Reviewer #4: Thank you for the opportunity to review this important study manuscript. Overall, as the first study of it’s kind in the Australian context, this is an important body of work in furthering the understanding of pattens of MC use and prescribing amongst clinicians here.

I have no major concerns and overall, would recommend the study for publication. I do however note one limitation that has largely not been addressed in the results or the discussion.

The study cohort comprises 546 people from the CAMS 20 dataset who report being prescribed MC products within the last year. The analyses rely on grouping these participants into 3 groups based on reported product composition. It is noted by the authors in the Methods section however, that 350 of these participants reported using both prescribed as well as illicit MC. The composition of the latter as acknowledged by the authors is hard to ascertain with clarity. It is not clear in the remainder of the manuscript how this issue has been dealt with in the reported results and the discussion. Further detail on this should be discussed and would place the results in context.

Reviewer #5: Abstract requires some additional information for clarity:

Clarification that those using CBD-dominant products were preferred by older individuals, non-inhaled and more females? What indications was CBD use reported for?

Where would balanced (1:1) products be considered?

What about people using multiple product types? There is a lack of detail on “efficacy” report measures, do you mean patient-reported effectiveness? When were these measured? Side effect profiles require quantification.

Drug interactions is mentioned in the conclusion but there is no data presented in the abstract to support this statement.

Article comments:

Some of the jurisdictional nuances re S4, S8 products require definition to improve clarity to an international audience

601 reported prescribed medical cannabis in the initial survey and 546 were included as a subgroup, if that means that 55 people were using illicit sources only? what is the rationale for excluding their experiences? If it was just that they did not report complete data mention that those without complete or unclear on product composition data were excluded.

How were patients using multiple cannabis products analysed?

The groups based on composition would be more useful if reclassified to align with the TGA cannabis product categories https://www.tga.gov.au/medicinal-cannabis-products-active-ingredients

The rationale for grouping TO and T+C is not well established, these patients could be taking balanced products with high amounts of CBD, as such contributing to both side effects and self-reported benefits.

The side effects table 3 is mostly THC related side effects so not seeing these with the CBD only group is logical. What would be more relevant and interesting would be to look at the differences between category 2-5 products – do side effect profiles change with CBD+THC vs THC alone?

Line 206-208 should be reworded as it currently states no differences but also differences. What are general presentations?

Table 6 – this data should be separated out THC only or THC+CBD? This data may be more clinically interesting re: potential entourage effects Was driving data different amongst acute and chronic exposures?

Line 277-280 should be reworded to exclude drug resistant epilepsy which has high quality evidence.

Minor:

If possible, add “in Australia” to the short title

Thorough editing as there are some extra periods and spacing in places

Reviewer #6: This is an interesting and well-written paper. The survey method and analysis are largely appropriate to the research questions being asked. Conclusions are stated in a careful and mostly balanced way. I enjoyed reading it. However, I do have some concerns about the survey design and analysis that ought to be addressed.

My main issues are outlined below:

1) Most of the subsample analysed here used both prescribed cannabis AND illegal cannabis for therapeutic/medicinal reasons, but I am not sure what measures referred to prescribed MC and can be attributed specifically to those products. The methods section does not specify e.g. if self-reported improvements in symptoms or side-effects were asked with explicit reference to prescribed MC? This may be a problem if a participant reported use of prescribed CBD-only, but also used THC from the illegal market... I would expect that the side effects/improvements would be different from CBD-only vs. if they answered the question considering ALL types of cannabis they used for therapeutic reasons. This also applies to other measures, e.g. driving after use of ANY product, or was the questions explicit and referred to driving after use of prescribed products?

2) Need more clarity about the definition of prescribed MC consumer. I.e. is ONE prescription in the past 12 or 6 months enough to qualify? Again, this is an issue as most participants in this analysis used both legal and illegal products; so it is possible they may be “mostly illegal MC consumers with only one MC prescription”. Did the survey asked whether they actually bought and used the prescribed product (or is receiving a prescription enough?). We know from other similar jurisdictions that some patients may request prescriptions simply to have it on hand in case of legal problems.

3) Route of administration (ROA) is mentioned in the analysis but not in the methods section. It would be good to get more clarity about the measures collected and which questions explicitly referred to prescribed MC. Regarding ROA, I can only see vaporizer and inhaler ROA in Table 1 – others not included?

The points above have significant implications for the conclusions we can draw from the analysis. I appreciate it is impossible to go back and ask more explicit questions linked to prescribed MC at this stage. The issues above ought to be addressed and controlled for, or discussed in the limitations. Overall, I enjoyed reading the study and I think this will be a useful contribution to the literature if authors are able to address the points above.

6. PLOS authors have the option to publish the peer review history of their article (what does this mean?). If published, this will include your full peer review and any attached files.

Reviewer #1: No

Reviewer #2: No

Reviewer #3: No

Reviewer #4: No

Reviewer #5: No

Reviewer #6: No

---

## [Author Response · Author response to Decision Letter 0]

16 Nov 2023

The Editor 

PLOS ONE

November 12, 2023 

Re: Revised manuscript – “Differences in prescribed medicinal cannabis use by cannabinoid product composition: Findings from the CAMS-20 Australia-wide study”

Dear Editor, 

Thank you for considering our manuscript, and for the opportunity to revise some aspects. I have addressed your feedback – in particular re patient content and patent declaration which I have added into the Methods (ethics) section and the competing interest section of the manuscript respectively. – This does not alter our adherence to PLOS ONE policies on sharing of data and/or materials. 

There are no ethical/legal restrictions on sharing the dataset so we will upload the minimal anonymous data set needed to replicate our study findings. It can be found at this website: https://osf.io/f2yc3/

We also thank the reviewers for their comments and suggestions, all of which we have considered and addressed. Our responses to the reviewers’ comments are attached.

Thank-you for considering our revised manuscript.  

  

Kind regards  

Dr Benjamin Trevitt

---

## [Decision Letter · Decision Letter 1]

27 Dec 2023

Differences in prescribed medicinal cannabis use by cannabinoid product composition: Findings from the cannabis as medicines survey 2020 (CAMS-20) Australia-wide study

PONE-D-23-23895R1

Dear Dr. Trevitt,

We’re pleased to inform you that your manuscript has been judged scientifically suitable for publication and will be formally accepted for publication once it meets all outstanding technical requirements.

Kind regards,

Andrea Mastinu

Academic Editor

PLOS ONE

Additional Editor Comments (optional):

Reviewers' comments:

Reviewer's Responses to Questions

**Comments to the Author**

1. If the authors have adequately addressed your comments raised in a previous round of review and you feel that this manuscript is now acceptable for publication, you may indicate that here to bypass the “Comments to the Author” section, enter your conflict of interest statement in the “Confidential to Editor” section, and submit your "Accept" recommendation.

Reviewer #1: All comments have been addressed

Reviewer #3: All comments have been addressed

Reviewer #4: All comments have been addressed

2. Is the manuscript technically sound, and do the data support the conclusions?

Reviewer #1: Yes

Reviewer #3: Yes

Reviewer #4: Yes

3. Has the statistical analysis been performed appropriately and rigorously? 

Reviewer #1: Yes

Reviewer #3: Yes

Reviewer #4: I Don't Know

4. Have the authors made all data underlying the findings in their manuscript fully available?

Reviewer #1: Yes

Reviewer #3: Yes

Reviewer #4: Yes

5. Is the manuscript presented in an intelligible fashion and written in standard English?

Reviewer #1: Yes

Reviewer #3: Yes

Reviewer #4: Yes

6. Review Comments to the Author

Reviewer #1: The authors have adequately addressed my comments and the comments of the other 5 reviewers; a quality article.

Reviewer #3: In the revised submission, the authors have adequately thoroughly my comments and suggested feedback. I have no remaining concerns.

Reviewer #4: Thank you for the opportunity to review this manuscript. The authors have addressed all the reviewer comments well and the manuscript is much stronger as a result. I would be happy to recommend acceptance of the is study for publication to the editor.

7. PLOS authors have the option to publish the peer review history of their article (what does this mean?). If published, this will include your full peer review and any attached files.

Reviewer #1: No

Reviewer #3: No

Reviewer #4: No

---

## [Editor Report · Acceptance letter]

18 Jan 2024

PONE-D-23-23895R1 

PLOS ONE

Dear Dr. Trevitt, 

I'm pleased to inform you that your manuscript has been deemed suitable for publication in PLOS ONE. Congratulations! Your manuscript is now being handed over to our production team.

Kind regards, 

on behalf of

Dr. Andrea Mastinu 

Academic Editor

PLOS ONE